# Effects of Vermicompost Substrates and Coconut Fibers Used against the Background of Various Biofertilizers on the Yields of *Cucumis melo* L. and *Solanum lycopersicum* L.

**Pedro A. Mejía** [1], **José Luis Ruíz-Zubiate** [2], **Amelia Correa-Bustos** [2], **María José López-López** [3] **and María del Carmen Salas-Sanjuán** [2,*]

1   PhD Program Protected Agriculture, Campus de Excelencia Internacional Agroalimentario, Almería University, ceiA3, La Cañada, 04120 Almería, Spain; pedromejia1205@gmail.com
2   Department of Agronomy, Campus de Excelencia Internacional Agroalimentario, ceiA3, Almería University, La Cañada, 04120 Almería, Spain; jl.ruizz@hotmail.com (J.L.R.-Z.); acb666@ual.es (A.C.-B.)
3   Department of Biology and Geology, Campus de Excelencia Internacional Agroalimentario, ceiA3, Almería University, La Cañada, 04120 Almería, Spain; mllopez@ual.es
*   Correspondence: csalas@ual.es

**Abstract:** Vermicompost has been promoted as a viable substrate component owing to its physico-chemical properties, nutrient richness, and status as an excellent soil improver. It is considered the best organic fertilizer and is more eco-friendly than chemical fertilizers. Plant-growth-promoting microorganisms (PGPMs) are defined as plant biofertilizers that improve nutritional efficiency—that is, they transform nutrients within substrates from organic to inorganic forms, making them available for plants. The main objective of this research study is to evaluate the effects of the application of three PGPM microbial consortia on different mixtures of organic substrates based on vermicompost (V) and coconut fiber (CF) on two different horticultural crops. We performed a yield analysis and drainage nutrient tests and determined the plant nutritional status and enzymatic activity in organic substrates based on the two crops, *Cucumis melo* L. and *Solanum lycopersicum* L. A multivariate analysis of variance and principal component analysis was conducted using substrate types and PGPMs as factors. Differences ($p < 0.05$) in yield, dehydrogenase activity, the nutrient concentrations in a petiole sap, and drainage were observed at 30, 60, 75, and 90 days after transplant. PGPMs such as *Trichoderma* sp. and plant-growth-promoting rhizobacteria (PGPR) in organic substrates (40V + 60CF) can significantly improve the nutritional status of plants for use in organic soilless container agriculture. Biofertilization with PGPMs and suitable mixtures of organic substrates together with aqueous extracts (tea) of vermicompost, as nutrient solutions applied by fertigation, has allowed us to achieve an adequate level of production through environmentally friendly techniques. The results obtained allowed us to affirm that it was possible to replace conventional fertilization using no chemical products and ensure adequate crop nutrition by supplying main macronutrients with organic sources and biofertilizers.

**Keywords:** plant growth promoting microorganisms; dehydrogenase activity; melon crop; organic substrates; soilless containers; tomato crop





## 1. Introduction

Soilless agriculture under plastic should focus on using new substrates that are efficient in terms of their profitability and do not harm natural resources, thereby facilitating sustainable agricultural production and reusing resources [1]. For this, it is necessary to identify alternative organic substrates that can support efficient and intensive crop production. The most important characteristics of a substrate are adequate porosity and readily available water and nutrients, which are essential for supporting plant growth [2].

Therefore, the growing medium must have a physical structure capable of maintaining a balance between air and water storage.

Sustainable practices for soilless container crop production, such as numerous organic materials that can be used as sources of nutrients and as substrates, are being applied more frequently in some crops (tomato, melon, lettuce). Organic substrates such as V and CF were appropriate for use in soilless containers [3]. Additionally, V as a substrate and aqueous extracts of the organic materials (tea) are viable alternatives to improve the production performance in soilless crops; this could be due to the presence of N-fixing and P-solubilizing bacteria, which were isolated from the gut of *Eisenia foetida* and improved the nutritional characteristics of the vermicompost [4,5].

Biofertilization with functional microorganisms, known as plant-growth-promoting microorganisms (PGPMs), increases the number of microorganisms in soil or substrate [6]. PGPMs can enhance plant growth and protect them from disease and abiotic stress through various mechanisms [6–8]. PGPMs can be naturally present in the root medium or inoculated during the crop cycle. A remarkable function of PGPMs is the improvement of the availability of nutrients for plants [6–8]. The biological fixation of nitrogen, the solubilization of phosphate, potassium, and nitrogen mineralization stand out [8]. This results in the acceleration of all microbial processes in converting the organic sources of nutrients into inorganic material available for plant uptake [6]. At the same time, the nutrient leaching and subsequent degradation of the agroecosystem are reduced [9,10]. Biofertilization has been demonstrated to be an excellent alternative to reducing the dependence on chemical fertilizers [11–13].

PGPMs are the main constituents of rhizospheric microbiota, which establish beneficial symbiotic relationships with plants through direct action mechanisms. In this way, PGPMs receive sugars (energy source) from the plant and in turn solubilize nutrients and micronutrients, produce growth regulators (hormones), suppress or control the production of stress hormones (ethylene), and improve water and nutrient consumption with the help of nitrogen-fixing microorganisms, phosphate-solubilizing bacteria, siderophore producers and others [14–16]. Bhardwaj et al. [17] reported that compared with conventional soil media, the use of organic substrates together with PGPMs (*Trichoderma* sp., arbuscular mycorrhizal fungi, and cabbage residues) showed superior results in terms of seed germination, root length, shoot length, root weight, shoot weight, and root/shoot ratio [17].

Increased interest in reducing chemical inputs in agriculture has led to the development of commercial biological inoculants to increase the mobilization of nutrients and enhance their availability to crop plants. In the context of progress in biotechnology relative to microorganisms and plants, the term plant biofertilizer (or PGPMs) is defined as substances or microorganisms applied to plants to improve nutritional efficiency and stress tolerance by abiotic factors and/or quality attributes of crops. Biofertilizers and microbial biostimulants (PGPMs), including mycorrhizae, fungi, and plant-growth-promoting bacteria, exert a dual function as biocontrol agents and biostimulants [18]. In this study, the effects of microorganisms on plant nutritional status were evaluated through a petiole sap test (SAP), which is a good indicator of plant nutritional status and probably a very effective method to diagnose plant nutrient deficiency and evaluate the effectiveness of fertility management [19].

This approach is considered a highly effective method. Other advantages include the possibility for early diagnosis of the nutritional potential for cultivation, which allows for adopting appropriate strategies to correct excessive or deficient nutrition programs of one or several essential nutrients. In addition, it allows alterations in the nutritional balance to be determined in real-time, which may be due to environmental effects or biotic and abiotic stresses, as reported by several authors [20]. Furthermore, these results are consistent with those reported by Ruiz and Salas [3]. They reported positive effects of PGPMs on atmospheric nitrogen fixation, ammonium oxidation, and increasing the assimilation of organic nutrients.

Arbuscular mycorrhizal fungi (AMF) and plant-growth-promoting rhizobacteria (PGPR) have been reported by Saia et al. [21] to be valuable options for farmers for improvements in yield, nutrient uptake, and agroecosystem sustainability. At the same time, *Trichoderma* sp. (TRICH) works as a biofertilizer and nutrient absorption enhancer [22–24]. PGPR shows greater functionality in substrates with high pH and salinity. Organic substrates based on mixtures of V and CF have physical-chemical and biological properties that allow for sustainable organic production using soilless cultivation, for which the hypothesis was established that the incorporation of PGPMs (bioaugmentation) and mixtures of organic substrates provide sufficient nutrients to plants (crops) to obtain acceptable yields.

Therefore, the main objective of this research was to evaluate the effects of different proportions (V + CF) of organic substrates and the application of biofertilizers (PGPMs) to provide a sufficient amount of nutrients to the crops to obtain an acceptable yield using soilless cultivation.

## 2. Materials and Methods

### 2.1. Plant Materials, Location, and Experimental Design

The research was conducted in a multi-tunnel type greenhouse with active climate control located on the Experimental Farm of the University of Almería (UAL-ANECOOP Foundation, 36.861905-2.282529, Retamar, Almería, Spain).

The research area was 300 m$^2$ for melon (*Cucumis melo* L.; Class Magnoliopsida, Order Cucurbitales, Family Cucurbitaceae) and tomato (*Solanum lycopersicum* L.; Class Magnoliopsida, Order Solanales, Family Solanaceae) crops in succession. Cultivation for both melon and tomato crops was carried out in polystyrene containers with a 27 L capacity. The entire process was completed in soilless media.

In general, the culture was carried out with different treatments of substrate mixtures during two production cycles, always using vermicompost tea as the main source of nutrients, with the application of exogenous PGPMs in some cases. The automated irrigation was based on dielectric sensors accounting for the percentage of humidity (volumetric water content) to minimize nutrient losses by leaching and maintaining optimal humidity. The percentage of humidity was changed if the percentage of drainage was modified.

The first crop was Galia melon variety Brisa (HM Clause seeds) at 1 plant m$^{-2}$, which was transplanted in March 2016, with the crop cycle finishing in June 2016. This was a short crop cycle in spring. The climatic conditions outside in these months are presented in Table 1.

**Table 1.** Temperature media (°C) outside the greenhouse in the melon crop.

| March | April | May | June |
|-------|-------|-----|------|
| 14.2 | 16.8 | 19.3 | 22.8 |

The second crop was the tomato variety Ramyle (Rijk Zwaan seeds) at 1.25 plant m$^{-2}$, transplanted in September 2016, with the crop cycle finishing in March 2017. This was a short crop cycle from autumn–winter. The climatic conditions outside in these months are presented in Table 2.

**Table 2.** Temperature media (°C) outside the greenhouse in tomato crop.

| September | October | November | December | January | February | March |
|-----------|---------|----------|----------|---------|----------|-------|
| 25.1 | 21.9 | 16.1 | 14.2 | 12 | 14 | 15 |

The experiment featured a randomized block design with four replicates of three containers (plants) per block. The treatments established depended on two factors, one of which was the type of PGPM applied, while the other was the volume percentages of V and CF in the substrate mixture. The handling, pruning, tutoring, and integral control of the crops were carried out according to the practices suggested for Almeria agriculture [25].

## 2.2. Description of Treatments: Substrates Used and PGPMs Applied

### 2.2.1. Substrates

The substrates used to prepare the mixtures in the test treatments were CF from Eji-turbas SLU- (El Ejido, Almería, Spain) and V from Tecomsa SLU-(Venta Gaspar, Almería, Spain). The physicochemical characterization of organic substrates evaluated in this research is described in Table 3.

**Table 3.** Physicochemical characterization of the initial coconut fiber (CF) and vermicompost (V) used for substrate mixtures (20V80CF (control), 40V60CF, 60V40CF) in the experiment.

| Parameters | Units | CF | V |
|---|---|---|---|
| pH | | 5.8–6.8 | 7.73 |
| Electric conductivity (E.C.) | $dS\,m^{-1}$ | <0.7 | 0.9 |
| Particle size | mm | 0–12 | <5 |
| Solid particle density | $g\,cm^{-3}$ | 0.1 | 0.77 |
| Total porosity | % | 95.4 | 67.9 |
| Cation exchange capacity | $mmol\,100\,g^{-1}$ | 60–130 | 25–30 |
| Organic matter | % DM [1] | 94.7 | 15 |
| Sodium | $mg\,L^{-1}$ | 253 | 431 |
| Potassium | $mg\,L^{-1}$ | 108 | 1881 |
| Calcium | $mg\,L^{-1}$ | 29 | 506 |
| Magnesium | $mg\,L^{-1}$ | 28 | 192 |
| Chloride | $mg\,L^{-1}$ | 281 | 1260 |
| Sulphates | $mg\,L^{-1}$ | 321 | 1988 |
| Nitrates | $mg\,L^{-1}$ | 120 | 1312 |
| Phosphates | $mg\,L^{-1}$ | 32 | 25 |
| Sodium adsorption ratio | | 11.4 | 4.1 |
| Organic carbon | $g\,kg^{-1}$ | 78.6 | 82.4 |
| Organic nitrogen | $g\,kg^{-1}$ | 1.8 | 9.9 |
| C: N Ratio | | 43.66 | 8.32 |
| Humic and Fulvic acids | % $w/w$ [2] | ND [3] | 17.50 |

[1] Dry matter; [2] weight/weight; [3] ND (not detected).

The substrate had three levels (number in front of the letter (V-CF) indicates the % by volume of the material in the mixture.): 20V80CF (control), 40V60CF, and 60V40CF (% $v/v$). Treatment 20V80CF was considered the relative substrate control treatment, as many studies have concluded that a rate of 20V80CF could be included without adversely affecting plant performance [26,27].

The physicochemical characterization was determined for each substrate treatment at the beginning. The samples were processed in a specialized and certified center for this analysis (Laboratorio Agroambiental FRAISORO UNE EN ISO 17025, Zizurkil, Gipuzkoa, Spain).

The analysis of organic material (OM) followed the methods of UNE-EN 13039: 2012. This method is the carbon fraction of a sample (5 g) free from water and inorganic substances, which is taken as equal to the loss on dry incineration at $(450 \pm 25)$ °C; the principle is that the test portion of the substrate (5 g) is dried at $(103 \pm 2)$ °C and then ashed at $(450 \pm 25)$ °C. The ash is determined as the residue on ignition. The OM is the loss of mass on the ignition, and both are expressed as a percentage by the mass of the dried sample.

The analysis of bulk density (BD), air volume (AV), total porosity (P), and readily available water (RAW) values followed the methods of UNE-EN 13041: 2012. The principle is that the sample is saturated in water and equilibrated on a sandbox at −50 cm water (−5 kPa) pressure head. The sample is then transferred into double-ring sample cylinders, rewetted, and equilibrated at −10-cm water (−1 kPa) pressure head. After equilibration, the physical properties are calculated (an equation for each property) from the wet and dry weights of the sample in the lower ring; it is also optional to apply the −50 and −100-cm water pressure heads, respectively.

The C/N ratio was calculated using gas chromatography and a thermal conductivity detector (PerkinElmer® EA2400, Waltham, MA, USA).

The concentrations of cations and anions, electric conductivity (E.C.), sodium adsorption ratio (SAR), and pH were determined in the saturated media extract (SME) prepared via ionic chromatography, i.e., nitrates ($NO_3^-$), chloride ($Cl^-$), sulfates ($SO_4^{2-}$), phosphates ($PO_4^{3-}$); atomic absorption spectrometry, i.e., calcium ($Ca^{2+}$), magnesium ($Mg^{2+}$), potassium ($K^+$), sodium ($Na^+$); electrometry, i.e., pH and E.C.; calculation (SAR); and a specific sensor electrode to ammonium ($NH_4^+$). The substrate samples were taken at a depth of 10 cm inside the container, and a paste was made using substrate and water (as an extracting solution) at a dilution ratio of 1:2. Then, the liquid portion was separated from the solid portion for pH, E.C., and main cation and anion analyses. All samples were processed in a lab certified to perform this type of test (Laboratorio Analítico Bioclínico LAB, UNE EN ISO/IEC 17025, PITA, Retamar, Almería, Spain).

### 2.2.2. Biofertilizers: Plant-Growth-Promoting Microorganisms

PGPMs were considered for different treatments, namely arbuscular mycorrhizal fungi (AMF) [28], plant-growth-promoting rhizobacteria (PGPR) [29], *Trichoderma* (TRICH) [30] and absolute control (no application of PGPMs).

The PGPMs used for the experiment had the following characteristics:

- **Arbuscular mycorrhizal fungi (AMF) consortium**: This treatment contained a mix of five different strains of mycorrhizal fungi at 5% (*w/w*) (equivalent to 150 spores $g^{-1}$) belonging to five species of the Glomeraceae family—*Glomus intraradices* (*Rhizophagus intraradices*, ID 4876), *Glomus deserticola* (*Septoglomus desertícola*, ID 1838035), *Glomus clarum* (*Rhizophagus clarus*, ID 94130), *Glomus mosseae* (*Funneliformis mosseae*, ID 27381) and *Glomus aggregatum* (ID 241619) [28]. These arbuscular mycorrhizal fungi (AMF) were isolated and extracted from the commercial product Bioradis Tablet (Bioera SL-Constantí, Tarragona, Spain). They were applied during three growth stages (transplant, flowering, and pre-harvest) in each crop using 1 g of AMF (containing 150 spores). This was introduced into the substrate (container) at a 10 cm depth in each plant [28] at each application. In the study, microbial activity was indirectly assessed through enzymatic activity and sap levels, but it was not assessed whether they achieved symbiosis with the plants in the experiment.
- **Plant-growth-promoting rhizobacteria (PGPR) consortium**: This treatment contained a mix of rhizobacteria, with $5 \times 10^9$ CFU (colony forming units) $g^{-1}$ of isolates of the Paenibacillaceae family—*Paenibacillus azotofixans* (*Paenibacillus durus*, ID ATCC 35681)—and Bacillaceae family—*Bacillus coagulans* (ID 941639) and *Bacillus pumilus* (ID ATCC 7061) [29]. Plant growth-promoting bacteria were isolated and extracted from the commercial product Bactel (Bioera SL-Spain). They were applied during three growth stages (transplant, flowering, and pre-harvest) in each crop at 100 mL from a suspension of 50 g $L^{-1}$ PGPR. We applied 100 mL to each plant (container) at 10 cm around the plant. This was equivalent to $2.5 \times 10^{10}$ CFU per plant [29] at each application.
- ***Trichoderma* sp. (TRICH) consortium**: This treatment contained a mix of *Trichoderma asperellum* (ID CBS 433.97) 0.5% g 100 $g^{-1}$ (*w/w*), $1 \times 10^8$ CFU $g^{-1}$ and *Trichoderma atroviride* (ID IMI206040) 0.5% *w/w*, $1 \times 10^8$ CFU $g^{-1}$ [30]. Isolates of Trichoderma fungi were isolated and extracted from the commercial product Tusal (CERTIS-Spain). These were applied during three growth stages (transplant, flowering, and pre-harvest) in each crop at 100 g TRICH concentration diluted in 1 $L^{-1}$ of water, and 100 mL of this mixture was applied to each plant (container) at 10 cm around the plant. This was equivalent to $10^9$ CFU plant$^{-1}$ [30] at each application.

To evaluate the total microbial load (bacteria and total fungi) and functional groups of the microbiota provided by each treatment consortium and vermicompost tea, the contents of total bacteria and fungi and the functional groups of nitrogen fixers (NF) and phosphate solubilizers (PS) were measured according to the following procedures. For the quantification of total bacteria (TB) and total fungi (TF), 0.1 mL of a suitable dilution was sown

in petri dishes with APHA (Panreac Química S.L.U., Barcelona, Spain) (TB) and Rose of Bengal (Panreac Química S.L.U., Barcelona, Spain) (TF) agar, respectively. The media were prepared following the manufacturer's instructions and distributed under aseptic conditions in sterile petri dishes measuring 9 cm in diameter. The sown volume was spread with sterile glass pearls, and after removing the pearls, the plates were incubated at 30 °C for 48 h (TB) and 5 days (TF). After the culture period, the colonies were counted, and the results were expressed in (colony forming units) CFU $g^{-1}$ of a solid sample or CFU $mL^{-1}$ of a liquid sample [31]. The quantification of P solubilizers followed the same procedure used for TB and TF, except that in this case the medium for phosphate solubilizers was tricalcium phosphate [32]. N fixers were also quantified using the plate colony count procedure. In this case, Burk agar medium without nitrogen was used [33], in which only those microorganisms capable of fixing atmospheric nitrogen ($N_2$) can survive. The sown media were incubated at 30 °C for 3–6 days, after which different colonial morphotypes were identified and counted. To verify the N fixers, a confirmatory test was performed so that each morphotype was isolated and sown again on a new Burk's N-free medium (BNF) plate. This operation made it possible to determine whether the growing microorganisms were $N_2$ fixers or used N fixed by other microorganisms. After incubation of the isolates at 30 °C for 24–48 h, a confirmatory reading was carried out in which the N-fixing morphotypes were verified (only those that grew in pure culture in BNF were considered N fixers). The result was expressed in CFU $g^{-1}$ dw or CFU $mL^{-1}$. Table 4 presents the initial mean values of the total microbial load (bacteria and total fungi) and the functional groups of the microbiota for each PGPM consortium, substrate, and vermicompost tea analyzed in this research study.

**Table 4.** Initial mean values of total microbial load (bacteria and total fungi) and functional groups of the microbiota provided by each PGPMs consortium (AMF, PGPR, and TRICH) and vermicompost tea used as an organic nutrient solution in fertigation (VT) evaluated in this research study.

| INPUTS | Units | TB * | TF * | NF * | PS * |
|---|---|---|---|---|---|
| AMF [1] | Log (CFU $g^{-1}$ dw **) | 5.88 ± 0.03 | 4.92 ± 0.08 | 5.62 ± 0.06 | 5.34 ± 0.13 |
| PGPR [2] | Log (CFU $g^{-1}$ dw) | 9.23 ± 0.31 | 3.77 ± 0.33 | 9.05 ± 0.27 | 0.00 |
| TRICH [3] | Log (CFU $g^{-1}$ dw) | 4.14 ± 0.29 | 7.65 ± 0.50 | 0.00 | 0.00 |
| VT | Log (CFU $mL^{-1}$ ***) | 4.93 ± 0.42 | 1.41 ± 0.33 | 4.83 ± 0.45 | 4.74 ± 0.06 |

[1] Arbuscular mycorrhizal fungi consortium. [2] Plant-growth-promoting rhizobacteria consortium. [3] Trichoderma consortium. * TB, total bacteria; TF, total fungus; NF, nitrogen fixers; PS, phosphate solubilizers. ** CFU $g^{-1}$ dw: colony-forming unit per gram dry weight. *** CFU $mL^{-1}$: colony-forming unit per milliliter.

## 2.3. Irrigation Management and Nutrition

A drip irrigation system was used with 4 L $h^{-1}$ flow drippers in each container. The main source of nutrients was vermicompost tea derived from horticultural vegetable waste [13,34] at 2.5 dS $m^{-1}$ of E.C., resulting in vermicompost aqueous extracts with the following concentrations in mmol·$L^{-1}$: $NO_3^-$ 1.5; $NH_4^+$ 0.3; $H_2PO_4^-$ 0.5; $K^+$ 4.8; $Ca^{2+}$ 2.3; $Mg^{2+}$ 1.3; $Cl^-$ 7.6; $Na^+$ 6.2 [35].

The pH control in the irrigation water was carried out to maintain a pH of 6.0–7.0 with acetic acid. Automated irrigation was established by determining the moisture available in the substrate using dielectric sensors installed in the containers (Decagon Pullman, Pullman, WA, USA), which also measured E.C. and temperature. The moisture content of the substrate was set across ranges of 20–30% and 10–15% of daily drainage in each treatment. For the production of VT, 5 tanks with a capacity of 1000 L each, a blower (1.0 kW, Motion Industries, Birmingham, AL, USA), aerating tubes (Jeneca, Chaozhou, China), and a 5 hp pump (C.R.I. group, Saravanampatty Coimbatore, India) were used. The V was mixed with irrigation water at a proportion of 0.1 Kg vermicompost solid $L^{-1}$ irrigation water, and the mixture was maintained with aeration for four days. Subsequently, the suspended solids were removed by filtration (0.125 mm), and the VT was injected into

the main irrigation pipe. The VT was the basis of the nutrient solution in fertigation, at an average injection ratio of 80 mL VT $L^{-1}$ of irrigation water [3].

The measurement of the percentage of drainage was calculated based on the volume of water drained between the volumes of irrigation applied daily to each treatment group. Three containers were selected to collect and measure daily drainage in each treatment to maintain the range established. Likewise, the contents of $NO_3^-$, $NH_4^+$, $Mg^{2+}$, $K^+$, $Cl^-$, $Na^+$, and $Ca^{2+}$ in the drainage collected at 30, 60, 75, and 90 DAT were evaluated in the control treatment group without PGPMs.

Petiole sap analysis is a method in which the sap is collected from a plant's leaves to evaluate nutrient levels, reflecting the nutrients immediately available to plants [36,37]. Samples of fresh petioles (20 for each treatment) were taken every three weeks, and petiole sap was extracted according to the plant petiole sap testing procedure for vegetable crops [38]. The concentrations of the main cations and anions were measured using anion and cation column chromatography. The time variable, days after transplant (DAT), was also taken into consideration in the statistical analyses of the parameters that were sampled throughout cultivation (sap and drainage analyses).

### 2.4. Crop Yield

During the experiment, the following measurement was taken to determine the effects of the treatments on crop yield. It was calculated in kg m$^{-2}$ based on fresh weight determined using a PCE-BS 3000 model balance with rank 3000 g, resolution 0.1 g, and precision ±0.3 g.

### 2.5. Dehydrogenase Activity (DHA)

During the experiment, to assess the microbiota activity in the rhizophore, dehydrogenase activity (IUBMB Enzyme Nomenclature 1.2.1.61) was determined in the substrate of each treatment [39,40], according to the protocol established by Casida [41]. This method is based on reducing 2,3,5-triphenyl tetrazolium chloride to triphenylformazan (TFF), whereby quantification is carried out using spectrophotometry (485 nm).

### 2.6. Data Analysis

The statistical analyses were performed using Statgraphics XVII-X64 software (Stat-Point, Inc., Herndon, VA, USA). Principal component analysis (PCA) was carried out to determine variability, correlations, and synergisms between different variables. A multivariate analysis of variance (ANOVA) was carried out using the Fisher's comparison test of means, expressing the statistically least significant difference (LSD) at $p < 0.05$. Further, we considered the dependence of some peculiarities of the vegetable crops' productivity (melon, tomatoes) on their water use, greenhouse climate, and crop cycle length to determine the appropriate statistical process [42].

The normality of the distribution of the evaluated parameters was tested for both crops based on Royston's H test at the 5% level of significance.

## 3. Results

### 3.1. Yield: PGPMs and Substrates in Each Crop (Melon, Tomatoes)

Table 5 shows the crop yields (melon, tomato) depending on the types of PGPMs used for biofertilization and substrate treatments. In the substrate factor of the melon crop, 40V60CF and 60V40CF presented the highest production ($p < 0.05$) rates with TRICH compared to the control substrate. In the tomato crop, only 40V60CF with TRICH stood out with the highest yield ($p < 0.05$).

The yields of plants treated with PGPMs, regardless of the type of substrate, significantly exceeded those of the controls. Among the various PGPMs treatments, TRICH significantly ($p < 0.05$) enhanced yields in substrates 40V60CF in tomato with higher levels than AMF, PGPR, and control treatments. In melon (Table 5), TRICH and AMF improved crop yields. Both factors exhibited significant differences ($p < 0.05$) in the interactions with each crop.

**Table 5.** Crop yield interactions between substrate treatments and PGPM treatments in the melon crop and the tomato crop.

| YIELD Kg m$^{-2}$ | Control | AMF | PGPR | TRICH |
|---|---|---|---|---|
| | | Melon | | |
| Control | 4.27 a A | 4.31 a A | 5.03 * B | 5.18 a B |
| 40V60CF | 4.75 b A | 5.43 b B | 4.98 * B | 5.51 b B |
| 60V40CF | 4.92 b A | 5.58 b B | 5.08 * A | 5.55 b B |
| | | Tomato | | |
| Control | 5.03 a A | 5.12 a A | 5.08 a A | 5.90 b B |
| 40V60CF | 5.74 b B | 5.96 ab B | 5.96 ab B | 6.12 b B |
| 60V40CF | 5.21 a A | 5.79 b B | 5.27 a A | 5.43 a A |

Substrate (% *v/v*) vermicompost (V) and coconut fiber (CF) (20V80CF (control), 40V60CF and 60V40CF). PGPMs (AMF (arbuscular mycorrhiza fungi), PGPR (plant-growth-promoting rhizobacteria), TRICH (*Trichoderma asperellum* + *Trichoderma atroviride*)); control (no PGPMs). Different small letters represent differences ($p < 0.05$) substrate treatments. Different capital letters represent differences ($p < 0.05$) among plant-growth-promoting microorganisms (PGPMs) among each substrate treatment. * No significant differences.

### 3.2. Petiole Sap Test

Tables 6 and 7 present the concentrations (mg L$^{-1}$) of the main cations and anions in the petiole sap of each treatment. The data are averages derived from three tests in the melon crop and five tests in the tomato crop. In general, the quantified concentration of NO$_3^-$ in petiole sap was low according to the references consulted, where NO$_3^-$ should have exceeded 1100 mg L$^{-1}$ in the initial stage of growth, and 700–800 mg L$^{-1}$ is acceptable in the final stage of pre-harvest [19,38,43]. In this research study, even with low levels of nitrates in the petiole sap, it was possible to obtain an acceptable level of production.

In the melon crop, there were significant differences ($p < 0.05$) in the levels of NO$_3^-$, NH$_4^+$, Mg$^{2+}$, Ca$^{2+}$, K$^+$, and Na$^+$ in sap for the type of substrate mixture among each of the PGPMs (Table 6, capital letters). In interrelation substrate treatments with PGPMs, PGPR and TRICH presented differences ($p < 0.05$), with higher levels of NO$_3^-$ in 40V60CF. Furthermore, NH$_4^+$ showed differences ($p < 0.05$) in TRICH and PGPR, with higher contents. Additionally, Mg$^{2+}$ showed differences ($p < 0.05$) in AMF and PGPR (lower level) as compared with TRICH and control. K$^+$ presented differences between control and PGPMs, higher in control ($p < 0.05$) than all other treatments. Ca$^{2+}$ also presented differences ($p < 0.05$), with higher levels in AMF and TRICH. Na$^+$ presented differences ($p < 0.05$), with lower concentrations found in control ($p < 0.05$).

Additionally, there were statistical differences ($p < 0.05$) caused by PGPMs among the different substrates (Table 6, small letters) in the levels of NO$_3^-$, NH$_4^+$, Na$^+$, K$^+$, Mg$^{2+}$, and Ca$^{2+}$. The differences showed that the higher proportions of V in treatments 40V60CF and 60V40CF were related to higher proportions of nutrients in the sap test.

In the tomato crop, there were no statistical differences ($p < 0.05$) between substrates treatments in the levels of NO$_3^-$, Mg$^{2+}$, Ca$^{2+}$, K$^+$, and Na$^+$ in sap among each of the PGPMs (Table 7, capital letters). In interrelated substrate treatments, the PGPMs presented differences, with higher levels ($p < 0.05$) of NO$_3^-$ than all other treatments. NH$_4^+$ did not show differences between PGPM treatments. Additionally, Mg$^{2+}$ showed differences ($p < 0.05$) between treatments, with higher levels of TRICH than control. K$^+$ did not present differences between treatments. Furthermore, Ca$^{2+}$ presented differences ($p < 0.05$), with higher levels of AMF and TRICH in 60V40CF. Na$^+$ presented differences ($p < 0.05$), with lower levels found in control ($p < 0.05$).

Additionally, there were statistical differences ($p < 0.05$) in PGPMs among the different substrates in the concentrations of NO$_3^-$, NH$_4^+$, Na$^+$, K$^+$, Mg$^{2+}$, and Ca$^{2+}$ in tomato sap (Table 7, small letters). The differences showed the same trend as the results for the melon crop. The greater proportions of vermicompost in the substrates 40V60CF and 60V40CF were related to increments in nutrients in the sap.

**Table 6.** Sap analysis (mg L$^{-1}$) of main cations and anions in the melon crop between substrate and PGPMs treatments.

|  | 20V80CF | | | | 40V60CF | | | | 60V40CF | | | |
|---|---|---|---|---|---|---|---|---|---|---|---|---|
|  | **Control** | **AMF** | **PGPR** | **TRICH** | **Control** | **AMF** | **PGPR** | **TRICH** | **Control** | **AMF** | **PGPR** | **TRICH** |
| $NO_3^-$ | 520.1 a A | 538.3 a A | 523.7 a A | 817.9 b B | 658.3 b B | 546.0 a A | 874.2 c D | 785.1 ab C | 488.7 a A | 683.7 b C | 593.2 b B | 767.3 a D |
| $NH_4^+$ | 108.3 a B | 98.3 a A | 100.3 a A | 105.2 a AB | 131.0 b A | 132.8 c A | 139.8 b B | 146.2 b B | 134.0 c B | 121.8 b A | 147.2 b C | 144.7 b C |
| $Mg^{2+}$ | 127.2 b C | 92.8 b B | 74.0 b A | 124.2 b C | 87.0 a B | 73.2 a A | 70.0 b A | 79.2 a AB | 120.0 b C | 73.2 a B | 56.8 a A | 134.0 b D |
| $K^+$ | 2018.8 a B | 1937.2 a AB | 1731.7 a A | 1864.1 a A | 2524.8 b C | 1752.0 a A | 2074.9 b B | 2092.8 b B | 2601.9 b C | 1921.7 a A | 2016.1 b AB | 2147.7 b B |
| $Ca^{2+}$ | 362.2 a A | 581.2 a C | 424.8 a B | 552.9 c C | 513.3 b C | 520.0 a C | 403.2 a A | 459.3 a B | 510.1 b B | 539.2a B | 425.1 a A | 555.2 b B |
| $Na^+$ | 184.0 b A | 331.7 a B | 343.2 ab B | 466.8 b C | 131.8 a A | 269.7 a B | 272.0a B | 301.2 a B | 163.0 b A | 259.2 a B | 308.7 b C | 296.8 a C |

Substrate (% *v/v*) vermicompost (V) and coconut fiber (CF) (20V80CF (control), 40V60CF and 60V40CF). PGPMs (AMF (arbuscular mycorrhiza fungi), PGPR (plant-growth-promoting rhizobacteria), TRICH (*Trichoderma asperellum + Trichoderma atroviride*); control (no PGPMs)). Different small letters represent differences ($p < 0.05$) substrate treatments. Different capital letters represent differences ($p < 0.05$) among plant-growth-promoting microorganisms (PGPMs) among each substrate treatment.

**Table 7.** Sap analysis (mg L$^{-1}$) of main cations and anions in the tomato crop between substrate treatments and PGPMs treatments.

|  | 20V80CF | | | | 40V60CF | | | | 60V40CF | | | |
|---|---|---|---|---|---|---|---|---|---|---|---|---|
|  | **Control** | **AMF** | **PGPR** | **TRICH** | **Control** | **AMF** | **PGPR** | **TRICH** | **Control** | **AMF** | **PGPR** | **TRICH** |
| $NO_3^-$ | 1138.6 a A | 1178.6 a A | 1365.6 a B | 1571.8 a C | 1392.7 c A | 1571.6 b B | 1600.3 b B | 1705.4 b B | 1070.0 b A | 1497.0 b C | 1298.8 b B | 1680.0 b D |
| $NH_4^+$ | 152.0 a * | 137.0 a * | 140.0 a * | 147.0 a * | 179.3 b * | 195.7 c * | 190.6 b * | 182.5 b * | 177.0 b * | 167.2 b * | 201.9 b * | 198.5 b * |
| $Mg^{2+}$ | 553.7 b C | 404.1 b B | 322.1 b A | 540.5 b C | 339.5 a B | 333.7 a B | 281.9 a A | 391.3 a C | 348.2 a A | 362.0 a A | 377.9 c A | 583.3 a B |
| $K^+$ | 3396.4 a B | 3259.1 a B | 2913.4 a A | 3136.2 a AB | 3520.8 ab AB | 3879.8 b C | 3566.9 b B | 3239.7 a A | 3704.6 b B | 3233.1 a A | 3476.0 b A | 3613.3 b AB |
| $Ca^{2+}$ | 608.7 a A | 976.8 c C | 713.9 a B | 929.3 b C | 641.1 a A | 778.7 a B | 945.3 b C | 821.0 a B | 857.3 b B | 906.2 b BC | 714.4 a A | 933.1 b C |
| $Na^+$ | 770.0 a A | 1387.9 b B | 1436.0 c B | 1953.5 c C | 940.6 ab A | 1041.8 a B | 1049.8 b B | 1046.2 a B | 1100.6 b B | 1084.5 a B | 873.2 a A | 1242.1 b B |

Substrate (% *v/v*) vermicompost (V) and coconut fiber (CF) (20V80CF (control), 40V60CF and 60V40CF). PGPMs (AMF (arbuscular mycorrhiza fungi), PGPR (plant-growth-promoting rhizobacteria), TRICH (*Trichoderma asperellum + Trichoderma atroviride*); Control (no PGPMs)). Different small letters represent differences ($p < 0.05$) substrate treatments. Different capital letters represent differences ($p < 0.05$) among plant-growth-promoting microorganisms (PGPMs) among each substrate treatment. * No significant differences.

### 3.3. Drainage Test

In addition to carrying out a sap analysis to monitor crop nutrition, the drainage was evaluated to assess and potentially avoid nutrient loss due to leaching. Table 8a shows the concentrations of $NO_3^-$, $NH_4^+$, $Mg^{2+}$, $K^+$, $Cl^-$, $Na^+$, and $Ca^{2+}$ in drainage samples from the melon crop, showing clear differences ($p < 0.05$) (E.C. and pH not shown as they did not show significant differences).

**Table 8.** (**a**) Concentrations of nutrients in drainage samples from the melon crop; (**b**) Concentrations of nutrients in drainage samples from the tomato crop.

| (a) | | | | | | | |
|---|---|---|---|---|---|---|---|
| Factor | $NO_3^-$-N mg L$^{-1}$ | $NH_4^+$ mg L$^{-1}$ | $Mg^{2+}$ mg L$^{-1}$ | $K^+$ mg L$^{-1}$ | $Cl^-$ mg L$^{-1}$ | $Na^+$ mg L$^{-1}$ | $Ca^{2+}$ mg L$^{-1}$ |
| **Substrate** | | | | | | | |
| 20V80CF | 1.76 a | 25.44 a | 29.65 a | 213.09 a | 255.62 a | 203.69 a | 298.18 a |
| 40V60CF | 1.30 a | 11.18 b | 42.78 a | 260.79 a | 394.59 ab | 269.67 ab | 376.73 ab |
| 60V40CF | 1.44 a | 11.37 b | 41.32 a | 270.95 a | 470.46 b | 297.03 b | 460.9 b |
| **DAT** | | | | | | | |
| 30 | 1.07 a | 17.14 b | 81.66 c | 467.23 a | 485.00 b | 231.97 ab | 368.72 a |
| 60 | 0.94 a | 9.02 a | 38.65 b | 199.4 b | 267.67 a | 217.25 a | 380.74 a |
| 75 | 2.27 b | 37.53 c | 13.61 a | 236.94 b | 412.67 b | 315.19 b | 416.01 a |
| 90 | 3.03 c | 12.27 ab | 25.28 ab | 255.31 b | 748.77 c | 420.71 c | 369.92 a |
| (b) | | | | | | | |
| Factor | $NO_3^-$-N mg L$^{-1}$ | $NH_4^+$ mg L$^{-1}$ | $Mg^{2+}$ mg L$^{-1}$ | $K^+$ mg L$^{-1}$ | $Cl^-$ mg L$^{-1}$ | $Na^+$ mg L$^{-1}$ | $Ca^{2+}$ mg L$^{-1}$ |
| **Substrate** | | | | | | | |
| 20V80CF | 1.63 b | 7.38 a | 8.02 a | 327.57 a | 229.36 a | 152.42 a | 69.60 b |
| 40V60CF | 4.48 a | 7.56 a | 11.66 a | 230.63 a | 199.58 a | 203.69 a | 119.60 a |
| 60V40CF | 3.83 a | 7.20 a | 9.48 a | 210.69 a | 263.39 a | 238.17 a | 144.00 a |
| **DAT** | | | | | | | |
| 30 | 2.71 b | 4.5 b | 0.24 b | 427.25 a | 263.39 bc | 78.16 c | 199.20 ab |
| 60 | 2.21 b | 10.98 a | 40.09 a | 392.46 a | 180.08 d | 222.54 b | 231.20 a |
| 90 | 5.85 a | 7.2 ab | 1.46 b | 345.94 a | 246.73 bcd | 159.78 b | 106.00 c |
| 120 | 5.12 a | 11.88 a | 0 b | 343.60 a | 341.38 a | 228.29 b | 131.60 bc |
| 150 | 8.57 a | 3.96 ab | 53.95 a | 87.17 b | 146.05 cd | 142.53 bc | 150.00 abc |
| 180 | 6.70 a | 1.8 b | 1.46 b | 58.24 b | 275.80 ab | 303.47 a | 110.00 c |

According to substrate mix (% *v/v*) of vermicompost (V) and coconut fiber (CF) (20V80CF (control), 40V60CF and 60V40CF) and days after transplant (DAT): 30, 60, 90, 120, 150 and 180. Different letters show significant differences—LSD Fisher 95%.

An evaluation of DAT levels in melon drainage samples presented significant differences in terms of $NO_3^-$, $Cl^-$, and $Na^+$, which presented the highest concentrations in DAT 90, while $NH_4^+$ and $Ca^{2+}$ presented their highest concentrations in DAT 75. In addition, the highest concentrations in DAT 30 were obtained for $Mg^{2+}$ and $K^+$.

Table 8b shows differences in concentrations of $NO_3^-$, $NH_4^+$, $Mg^{2+}$, $K^+$, $Cl^-$, $Na^+$, and $Ca^{2+}$ in drainage samples from the tomato crop ($p < 0.05$) (E.C. and pH not shown as they did not show significant differences). The evaluation of DAT levels in the tomato crop showed differences, with $NO_3^-$ especially presenting an increasing trend over time, in contrast to $K^+$, which showed a decreasing trend over time. The cause of the increase in $NO_3^-$ may have been due to the degradation that the substrate undergoes over time by the PGPM, thereby increasing the concentration of nutrients available to the plant.

### 3.4. Dehydrogenase Activity (DHA)

Figure 1a presents the results of the DHA in the melon crop between substrate treatments and PGPMs (AMF, PGPR, TRICH, and control). Regarding the substrate factor, the 40V60CF and control substrate presented the highest DHA activity levels compared to 60V40CF, even considering that melon has a short production cycle (three months).

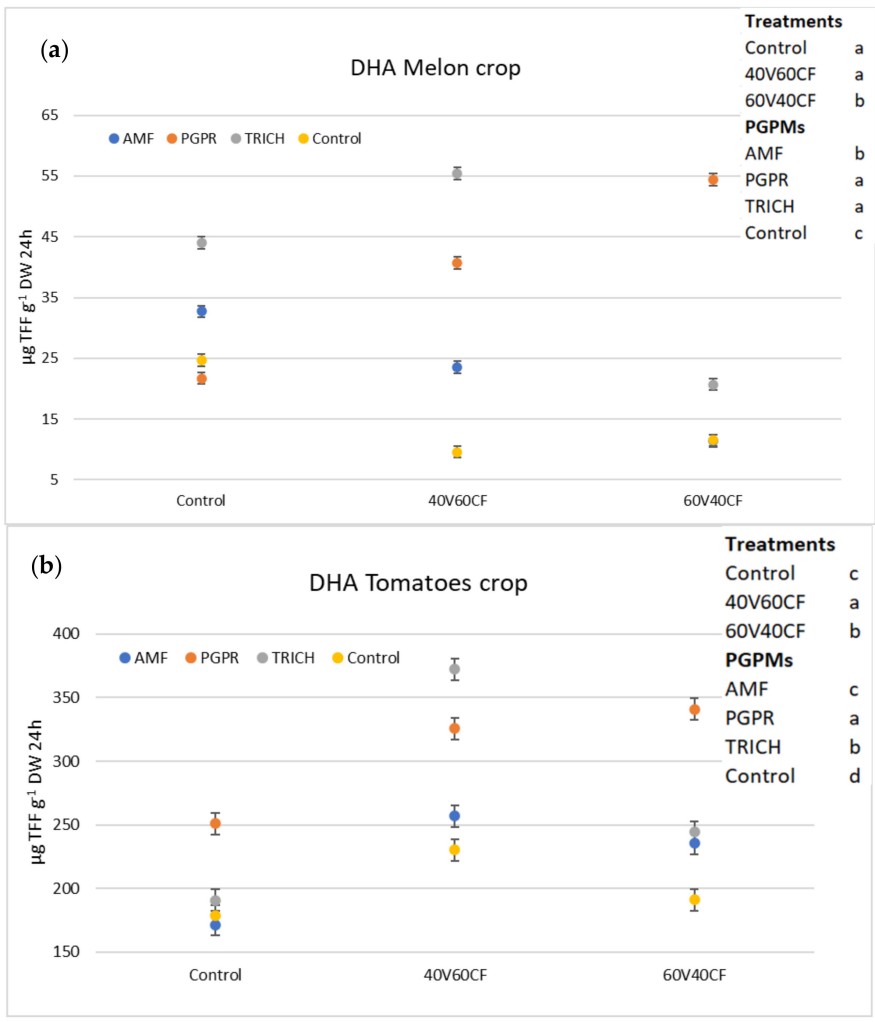

**Figure 1.** Dehydrogenase activity (DHA) between substrate treatments and PGPMs in (**a**) melon crop and (**b**) tomato crop. Substrates (% *v/v*) of vermicompost (V) and coconut fiber (CF) (20V80CF (control), 40V60CF and 60V40CF). PGPMs (AMF (arbuscular mycorrhiza fungi), PGPR (plant-growth-promoting rhizobacteria), TRICH (*Trichoderma asperellum* + *Trichoderma atroviride*) and control (no PGPMs). Different letters in substrate legend represent differences ($p < 0.05$) among substrate treatments. Different letters in PGPM legend represent differences ($p < 0.05$) among plant-growth-promoting microorganisms (PGPMs).

Figure 1b presents the results of the DHA in the tomato crop. The results for the substrate factor indicate that 40V60CF presented the highest DHA activity, while the control substrate presented the lowest activity. Both factors (substrate and PGPMs) showed significant differences ($p < 0.05$) in interactions in each crop.

### 3.5. Principal Component Analysis

The PCA was carried out to describe most of the present variability and to find nutritional parameters (cations and anions) that could work synergistically in terms of improving yield and DHA. One criterion for selecting the number of main components to

be extracted is to select all components for which the corresponding eigenvalue is at least one (fraction 1/p of the total population variance).

Figure 2a presents the results of the melon crop PCA carried out on cations, anions, yields, DHA, substrates, and PGPMs, showing that three main components were selected, with the first and second components accounting for 55.76% of the total variance. The variables in component 1 showed a direct correlation and synergy between yield and DHA, with $NO_3^-$ linked to TRICH and PGPR in substrates with a higher amount of vermicompost (40V60CF, 60V40CF), also indicating that yield is correlated with higher PGPM activity. This can be interpreted by the potential of microorganisms associated with organic matter to increase enzyme activity (DHA) and consequently improve the availability of $NO_3^-$ and $NH_4^+$ in response to the increased availability of N due to the activity of microorganisms and increased crop yields. On the other hand, the application of AMF did not influence PC1. It can also be seen that the variables that most influenced the variability (greatest distance from the midway point) were $K^+$, $NH_4^+$, $Na^+$, and $NO_3^-$, in contrast to $Mg^{2+}$ and $Ca^{2+}$.

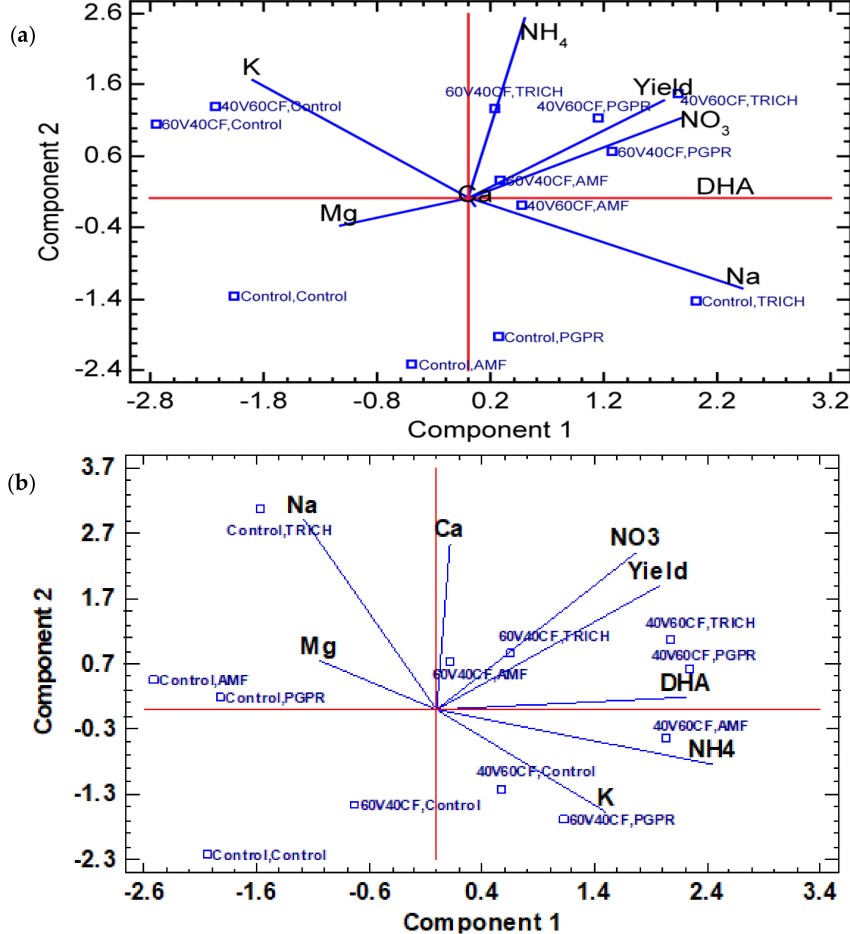

**Figure 2.** Principal component analysis bi-plot of component weights between the main cations and anions. (**a**) Sap and yield of the melon crop according to substrates and PGPMs treatments. First component (31.64% variance), second component (24.12% variance). Substrate treatments (% *v/v*): vermicompost (V) and coconut fiber (CF) (20V80CF (control), 40V60CF and 60V40CF as indicated. PGPMs treatments were AMF (arbuscular mycorrhiza fungi), PGPR (plant-growth-promoting rhizobacteria), TRICH (*Trichoderma asperellum + Trichoderma atroviride*), and control (no PGPMs), as indicated. (**b**). Sap and yield of the tomato crop according to the substrates and PGPM treatments. First component (37.31% variance), second component (27.38% variance). Substrate treatments (% *v/v*) vermicompost (V) and coconut fiber (CF): 20V80CF (control), 40V60CF and 60V40CF. PGPMs treatments were AMF (arbuscular mycorrhiza fungi), PGPR (plant-growth-promoting rhizobacteria), TRICH (*Trichoderma asperellum + Trichoderma atroviride*), and control (no PGPM), as indicated.

Figure 2b presents the results of the tomato crop PCA carried out on cations, anions, yields, DHA, substrates, and PGPMs, in addition to determining the three main components, with the first two accounting for 64.69% of the total variance. In the variables in component 1, as with the melon crop, there was a direct correlation and synergy between DHA, $NO_3^-$ and yield. In this case, the yield was signally linked to TRICH, while DHA was strongly linked to PGPR. The yield was strongly linked to the substrates 40V60 CF and 60V40CF in the area of greatest activity of PGPMs (TRICH, PGPR, and AMF). The variables that most influenced data variability were all of the analyzed ions except for $Mg^{2+}$ and $Ca^{2+}$. It is also possible to consider the controls independently of the enzymatic activity, which demonstrates that the PGPMs acted as biofertilizers and the main activators of the enzymatic activity in the rhizosphere microbiota.

## 4. Discussion

The results of this study showed that a mixture of V and CF can be used in soilless crops as a source of nutrients and as substrates in organic crop production. These results coincide with those obtained by several authors who have stated that the higher the proportion of vermicompost, the more cations and anions are released [44], meaning the plants consequently will have more nutrients available for their growth and production compared to using only CF as a substrate.

DHA showed similar trends between the two crops, although the level was higher in tomato plants, potentially because the rhizosphere microbiota had more time (due to the longer cycle) to increase its population dynamics, making the transformation from organic to inorganic sources available for plants. This suggests that the reduction in DHA in the melon crop may have been because the microbiota of the substrate had a prolonged adaptation phase (lag) and did not reach its maximum growth, resulting in reduced activity. However, the tomato crop (cycle of six months of cultivation) achieved maximum development and showed higher DHA. Furthermore, lower levels of DHA were seen in control without PGPMs in both crops, tomato, and melon. The enzymatic activity, DHA, allows an assessment of the microbial activity.

The results of the petiole sap test for different substrates showed that there was variation in nutrient concentrations. Interactions between factors substrate and PGPMs in both crops showed that TRICH and PGPR were related to higher levels of $NO_3^-$, which could be related to the higher concentration of bacteria NF in PGPR (Table 4). TRICH and AMF presented higher levels of $Ca_2^+$ and lower levels of $Na^+$ in control without PGPMs. These levels of nutrients were inside the ranges established by several authors [45]. PGPR and TRICH presented higher levels of $NO_3^-$ when they were applied to the mixture of substrates 40V60CF and 60V40CF in the two crops. The nutritional richness present in V, reported by several authors, is fully available to plants if they have an adequate balance or percentage in their ability to exchange cations [46]. This was also reported by Wang et al. [47,48], who evaluated the productivity, quality, and levels of $NO_3^-$ and $NH_4^+$ between conventional and compost treatments and concluded that vermicompost presented the highest concentrations of $NO_3^-$ and $NH_4^+$ of the evaluated treatments, so it is recommended as a biofertilizer for crops.

The drainage analysis concentrations of $NO_3^-$, $K^+$, $Na^+$, and $Ca_2^+$ increased in the substrate mixtures with higher amounts of vermicompost and in relation to the DAT during the crop cycle. The concentrations were initially low; however, the microbial activity (DHA) increased, meaning the availability of nutrients increased throughout the crop cycle. Higher availability coincided with the higher nutritional demands of the crop during the growth and development of fruits. The increase in DHA is related to the activity of the rhizosphere microbiota at its highest growth potential, as reported by Vargas-Garcia et al. [49]. $NO_3^-$ is one of the nutrients that tend to be limited when carrying out fertilization with organic sources in organic farming. Therefore, these results could be of great importance considering that $NO_3^-$ concentrations were higher for all the evaluated PGPMs compared to the control ($p < 0.05$). Additionally, according to the United States

Environmental Protection Agency (EPA), levels of nitrate-N (Table 8) at or below 10 mg $L^{-1}$ are considered safe for everyone [50]; in this research, the level of nitrate-N was below this standard.

However, it is important to remember that in this research study, fertigation was applied using vermicompost tea in all irrigations [13], which allowed the nutritional status of the crops to be kept and adequate production levels to be obtained. Relying only on the nutrients of the substrate would not allow sufficiently high-yielding production to be achieved for horticultural crops grown without soil.

One of the main sources of richness in the vermicompost tea + organic substrate is the high potential of nitrogen-fixing, phosphate solubilizing bacteria, and siderophore-producing bacteria. Mal et al. (2021) [51] reported that the integration of nitrogen-fixing and phosphate solubilizing bacteria with the vermicomposting process resulted in substantial enrichment of the product and significant improvements in the population of the inoculated microorganisms, making it a potential biofertilizer.

Additionally, it had the best nutritional status in biofertilized plants, providing evidence that microorganisms can facilitate the availability and assimilation of some nutrients ($p < 0.05$) for plants and can be used to ensure profitable and sustainable production without using chemical fertilizers. Both factors, the sap analysis and DHA, showed significant differences ($p < 0.05$) in the interactions for each crop. Microorganisms are present in organic substrates, which work to convert organic nutrients into assimilable forms for plants. Referring specifically to the PGPMs evaluated in this research study, *Trichoderma* (*T. asperellum* + *T. atroviride*), a eukaryotic organism, had no direct influence on atmospheric nitrogen fixation or ammonium oxidation.

However, there are several reports on the indirect stimulation of plant growth through different functionalities. The results obtained are in line with those obtained by Domínguez et al. [52] and Harman et al. [53]. *Trichoderma* sp. has also been reported as a plant growth stimulant, even under saline conditions [54] and water stress. It has also been noted to increase carbon and nitrogen levels in plants [55]. Pascale et al. [56] reported that *Trichoderma* sp. and its secondary metabolites improve the yield and quality of grapes. Moreover, the ability of *Trichoderma atroviride* to utilize plant waste byproducts was tested by Matata et al. [57]. They suggested that high molecular weight proteases may facilitate the heterotrophic–saprophytic mode of life of this fungus. The results of that study suggest that *T. asperellum* and *T. atroviride*, when applied to organic substrates, improve their adaptability and mode of action, which enhances nutrient availability in the plant rhizosphere. In addition, Ruting et al. [58] stated that xylanases present in *Trichoderma asperellum* promote growth and enhance the stress resistance of plants. Muniswami et al. [59] demonstrated the application of *Trichoderma* as a biofertilizer for maize.

Additionally, regarding TRICH and PGPR, both microorganisms have been reported as plant growth promoters and photosynthesis stimulants; an N-fixing bacterium ($N_2$) [4], an ammonia-oxidizing bacterium, and an ammonia-oxidizing archaea have been reported as components of PGPR [60]. Notably, in the substrate with the highest amount of V, treatment with PGPR led to the highest level of DHA, suggesting that there is a direct relationship whereby the higher the vermicompost content, the higher the DHA, possibly due to increases in the concentrations of nitrifying and ammonifying bacteria following the exogenous application of PGPR, as reported by Lang and Elliot [61]. In their paper, they indicated that inoculation with nitrifying bacteria may assist in the overall management of N in the rhizosphere and may be a feasible alternative for preventing either $NH_4^+$ or $NO_2$ phytotoxicity. A higher DHA with a higher proportion of V in the substrate mixture suggests that the composition of the organic matter can contribute to increasing microbial activity.

The results suggested PGPRs excelled above all other treatments in both crops because they presented higher activity for the substrate 60V40CF. PGPRs presented better adaptation and performed more activities with a higher percentage of vermicompost. This could also be related to the findings published by Tao et al. [60], who stated that ammonia-oxidizing

bacteria are more suited to different fertilization regimens in calcareous or saline soils, similar environments to that for 60V40CF, with a higher concentration of cations due to its higher V content (see Table 6). Both factors (substrate and PGPMs) showed significant differences ($p < 0.05$) in interactions for each crop.

It is also important to highlight from the results of this work that with the activity of the PGPMs, the necessary concentrations of nitrates in the sap can be obtained, facilitating adequate production. It is possible to grow organic soilless crops by combining PGPMs and organic substrates (V + CF), meaning an environmentally friendly production system can be achieved without chemical fertilizers. This was also reported by Djukic et al. [62], whereby the application of biofertilizers led to a higher yield in potatoes compared to the application of chemical fertilizers.

Therefore, according to the results, the 40V60CF substrate behaves better than the other substrates (control, 60V40CF) as a source of nutrients, which are solubilized and mineralized (nitrates) by microorganisms (PGPMs) from organic matter. In the mixture, 40V60CF allows the maximum efficiency of PGPMs in terms of the conversion process from organic to inorganic forms available as nutrients for plants, according to the PCA analysis where DHA, $NO_3^-$ and yields appeared together in the same component (Figure 2). The mix of a substrate with PGPMs coincides with the result reported by Das et al. [44]. They reported that the combination of vermicompost with PGPMs (*Trichoderma viride* (cellu-lolytic), *Azotobacter chroococcum* (N-fixer), and *Bacillus polymyxa* (P solubilizing) enriched the nutritional content of the substrate.

## 5. Conclusions

There were differences ($p < 0.05$) in the soilless crop production rates of melon and tomato according to the biofertilizer (PGPMs) applied to the organic substrate (V + CF) compared to the control (without application of PGPMs). By applying *Trichoderma* sp. consortium sp. and PGPR (plant-growth-promoting rhizobacteria consortium), high yields were obtained for the different organic substrates ($p < 0.05$) in the substrate treatments that contained the highest percentages of vermicompost (40V60CF and 60V40CF). The 40V60CF substrate showed the best nutritional balance between cations and anions in the sap test compared to the control substrate and 60V40CF.

The increase in the proportion of vermicompost in the substrate mixture and the exogenous contribution of PGPMs allowed us to obtain higher yields. To achieve increased productivity, it is recommended that when organic substrates (V + CF) are used, microbial biofertilizers should also be used since they accelerate the transformation process and the availability of nutrients such as nitrate with low concentrations in organic materials.

Biofertilization with PGPMs and adequate mixtures of organic substrates together with vermicompost tea as a nutrient solution applied by fertigation allowed us to achieve environmentally friendly production with a low nitrate-N concentration when compared to the standard established for drainage in soilless crops.

The highest nutrient concentrations in the drainage were present in the mixtures with the highest volumes of vermicompost (40V60CF and 60V40CF), coinciding with the treatments with higher production and DHA activity.

When using PGPMs in soilless cultivation, it is crucial to consider the physicochemical and biological properties of the substrate.

**Author Contributions:** Data curation, P.A.M., J.L.R.-Z. and A.C.-B.; Formal analysis, P.A.M., J.L.R.-Z., M.J.L.-L. and M.d.C.S.-S.; Funding acquisition, P.A.M.; Investigation, P.A.M., J.L.R.-Z. and M.d.C.S.-S.; Methodology, A.C.-B., M.J.L.-L. and M.d.C.S.-S.; Supervision, M.J.L.-L. and M.d.C.S.-S.; Writing—original draft, P.A.M. and A.C.-B.; Writing—review & editing, P.A.M. and M.d.C.S.-S. All authors have read and agreed to the published version of the manuscript.

**Funding:** This research received no external funding.

**Institutional Review Board Statement:** Not applicable.

**Informed Consent Statement:** Not applicable.

**Data Availability Statement:** Not applicable.

**Conflicts of Interest:** The authors declare no conflict of interest.

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
