# Peer review of "Effects of Vermicompost Substrates and Coconut Fibers Used against the Background of Various Biofertilizers on the Yields of Cucumis melo L. and Solanum lycopersicum L."

_horticulturae, doi:10.3390/horticulturae8050445_

Round 1

Reviewer 1 Report

I rate the research topic undertaken by the authors very highly.

 The use of water extracts of vermicomposts together with bioactive microorganisms (microbial bio-fertilizers) may increase the yield of plants.

Notes on the manuscript:

line 1-1: I propose to correct the title: Effect of vermicompost substrates and coconut fibers used against the background of various bio-fertilizers on the yield of Cucumis melo L. and Solanum lycopersicum L.

line 34-36. I suggest keyword correction : active microorganisms, dehydrogenase activity, melon crop,  organic substrates, soilless containers. omato crop

line 377-385, table 3 and 4, line 397-398, table 5.

Please shorten the signature. I propose to put the description of the table directly below the table.

Author Response

Thank you very much for your advice, they have been taken into consideration and have been adapted to the content of the manuscript. We would like inform you that the manuscript was undergone English language editing by MDPI.

Kind regards

Reviewer 2 Report

In this study the authors used various combinations of soilless media (vermicompost and coconut fiber) and biofertilizer (three microbial consortiums) to determine the effect on melon and tomato yield. In additional to crop yield, nutrient analysis of drainage water and petiole sap was also determined. Results were analyzed with multivariate statistics, including MANOVA and PCA. The authors found that the biofertilizers did affect crop yield and nutritional status and conclude that their methods affirm that using environmentally friendly biofertilizers can replace the use of chemical fertilizers.

There are a few concerns I have with this paper, mainly the statistical analysis. The study is presented as a time dependent study and therefore a repeated measured analysis should be utilized. Time is clearly a factor in the results, but is not properly acknowledged in the statistical analysis.  I also think the figures could be redesigned. In Figure 1 for example, the significant letters should be over the bars on the graph, not on a separate table in the graphic. It was hard to “see” the significance of the results. Finally, certain aspects of the paper were difficult to read. Another round of editing is necessary.

Specific line comments:

 L38-40: This is a very broad statement that current agriculture should focus on using soilless substrates and I think it needs to be qualified as those crops that aren’t grown in the field. It isn’t feasible to grow all of the world’s food in without soil.

L42: I’m unsure of what “integral” means in this context.

L48. Some crops are mentioned, can you give an example of some?

L48-49: What were V and CF demonstrated to be appropriate for?  

L65-66: In the sentence, “Biofertilization is…on these” – what is the “these” referring to?

L87: space needed after [15].

L88: unnecessary space after (SAP)

L106: delete “vermicompost”, it was already introduced so V is all that is needed.

L107: delete “coconut fiber”, it was already introduced so CF is all that is needed.

L118: capitalize Design

L125: rearrange, put 27 L before capacity

L126: delete “each one”

L127-131: This sentence was hard to follow.

L130-131 and 135-137: The temperature data might be easier to read in a table.

L141: rearrange end of sentence. Instead of in the mixture of substrate, use substrate mixture.

L145-146. Very confusing sentence “The physicochemical characterization of initial it was described by enterprise.” I also don’t know what enterprise is.

L146: capitalize table.

L176: put the location after the company and model.

L189-191: Strange wording. Add “different” between four and treatments. The “i.e.” isn’t really appropriate for this sentence – maybe a colon would be more appropriate.

L201. Change “There” to They

L203. Rewrite. Put “containing 150 spores” in parentheses and end the sentence there. Then start the new sentence with “It”

L216. End the sentence after PGPR and start the new sentence with “It”

L218. How did you apply this? The AMF was very specific at 10 cm depth.

L227. How did you apply this? The AMF was very specific at 10 cm depth.

L233: What is the location of Panreac?

L242: delete “was used”

L244: put N2 in parentheses.

L249: change nitrogen to N

L251: change nitrogen to N

Table 2: What about the background load of the control?

L273: After Decagon, put Pullman, WA USA.

L276: put brand and location after the blower and the pump

L277: change Vermicompost to V

L285: do you mean established instead of stablished?

L288-295: How many petioles were measured per plant?

L302: confusing sentence

Figure 1: Put the letters above the bars so the reader can see the differences.

L356. Confusing sentence

L356. Don’t forget the charge on the NO3. This happens a few more times throughout.

L360-361: Very confusing sentence “Besides NH4+ not showed differences between PGPMs treatments.” Besides doesn’t seem to be the correct word to use in this situation. You should also put a “did” in from of the “not”

L362: extra space between p and <0.05

L363: extra space between p and <0.05

L364: confusing sentence. Unsure of what “about Na+ presented differences…” means.

L368: Move the table reference to the end of the sentence.

L373-374: Unsure of what is being said with “than all other treatments”

L375: This sentence should be in the discussion and not the results.

L389-392: Tell us the results. Even though they are in the table you still should tell the reader what happened and reference the table.

Table 5: This is for the melon crop, what about the tomato crop?

L408-410: Where any of these things significantly different?

L411-419: This seems more appropriate for the discussion and not the results.

L427: Why is melon crop in parentheses? Instead, you could write Figure 3a present the result of the melon crop PCA.

L435. Add “and” between microorganisms and crop yields

L442: Besides seems out of place in this sentence.

L445. Add “the” between in and melon.

L446. Change to “microorganisms in the substrate”

L458: Put an “a” between as and source.

L462: change coconut fiber to CF

L463-470. These are results. Don’t just repeat the results, but discuss why.

L474: Besides is out of place in this sentence.

L488: change le to the

L548: confusing sentence

Author Response

(The authors gave the same response as above.)

Reviewer 3 Report

I have read this manuscript with a good deal of interest. This work is of immense importance to showcase the potential of PGPMs and organic substrate for the production of soilless crops. The results of this research can be considered relevant from the points of view of environmental resource protection and food security. I found this MS acceptable for publication prior necessary amendments, these are

The authors should better develop a hypothesis. 

Lines 55-62, need to cite suitable 2 or more refernces preferably review articles. I provide some of the most relevant here (The good, the bad, and the ugly of rhizosphere microbiome. In Probiotics and plant health (pp. 253-290). Springer, Singapore)

(Insights into the interactions among roots, rhizosphere, and rhizobacteria for improving plant growth and tolerance to abiotic stresses: A review)

Rhizosphere Bacteria in Plant Growth Promotion, Biocontrol, and Bioremediation of Contaminated Sites: A Comprehensive Review of Effects and Mechanisms

The details on how the plants were raised is missing. How The necesary growth conditions and agronomic requirements of the crop were met during experiment should be provided.

Did the authors also checked the consortium application of the applied microbes? 

Did the authors also checked the final CFU of PGPR after applicationn during or at the end of the experiment?

some of the odd expressions are there which reduce the quality of the MS to be accepted lines 302-3030 for example. 

AFM OR AMF?

What is the take home message?itations and implications of your study in the conclusion section.

Author Response

(The authors gave the same response as above.)

Round 2

Reviewer 2 Report

L299: stablished should be changed to established

Reviewer 3 Report

The authors have done a good job. All the comments have been incorporated. The MS is ready for acceptance.

This manuscript is a resubmission of an earlier submission. The following is a list of the peer review reports and author responses from that submission.

Round 1

Reviewer 1 Report

The experimental work presented in the Manuscript horticulturae-1559742, entitled "Organic substrates and biofertilization in soilless crops and their effects on yield" is an interesting research related with the utilization of organic materials for growing media and microorganisms as biofertilizers for soilless crops. The article reports the effects of both substrates and microorganism on several variables and indicators in plants and substrate.   Despite It is a well-developed approach with some promising results, there are some shortcomings that should be considered in order to enhance the final manuscript. First, English need some editing, especially at the introduction section. 
Material and methods section was not adequately described (e.g., drainage test was poorly mentioned in this section) and also need to be improved. The experiment design was not really clear or appropriate. Author suggested a “randomized complete block design” under controlled conditions, and the control treatment could be not properly which is based on the initial mixture at a lower level for vermicompost (20%) using as reference a previous research where composted green waste and not vermicompost that was mixed with peat. Moreover, in the same section no references were provided for some critical methods as the application dose for microorganisms. 
The discussion section also need some improvements with statements included previously on results section. Therefore, the manuscript must be handled back to the authors for major revisions or suggestions for being published.

Author Response

Thank you so much about the review´s commentaries. We tried to modifiy all of them. We hope it is everything ok and to have enhanced the quality to the article.

Review 
•    English need some editing, especially at the introduction section. 
The paper has been editied in different lines. The changes:
o    The abstract lines are 23-25
o    Introduction lines are 40-45 and the moved information are 63-119.
o    Added a hypothesis in lines 120-125.
•    Material and methods section was not adequately described (e.g., drainage test was poorly mentioned in this section) and also need to be improved. The experiment design was not really clear or appropriate. Author suggested a “randomized complete block design” under controlled conditions, and the control treatment could be not properly which is based on the initial mixture at a lower level for vermicompost (20%) using as reference a previous research where composted green waste and not vermicompost that was mixed with peat. 
Material and methods were improved, in the case of experimental was corrected a mistake in line 134 and added more information about that in line 134-137. About the control level the vermicompost, it was changed reference previous research for another in line 143. This reference previous research is:
Prasad, M.; Maher, M. J. The use of composted green waste (CGW) as a growing medium component. Acta Hortic. 2001, 549, 107–114.
Truong, H. D., & Wang, C. H. (2015). Studies on the effects of vermicompost on physicochemical properties and growth of two tomato varieties under greenhouse conditions. Communications in Soil Science and Plant Analysis, 46(12), 1494-1506. 

The drainage test is mentioned in lines 238-242. At the same time, we added some recommendations of reviewer 2. This changes was in lines 219-228 ( with information about the culture). And subsection 2.5. was changed and added lines 247-248, 254-256 and 278-281.

•    Moreover, in the same section no references were provided for some critical methods as the application dose for microorganisms. 
Add cited in different parts of Material and methods with a number reference 32, 33 and 34, lines 144-146 and 163,168 and 171.

Ramadhani, I; Suliasih; Widawati, S.; Sudiana, I. M.; Kobayashi, M."INTERNATIONAL SYMPOSIUM ON BIOREMEDIATION, BIOMATERIAL, REVEGETATION, AND CONSERVATION". En: IOP Conference Series-Earth and Environmental Science. 308, 01/01/2019. ISSN 1755-1307.
Basavesha, K. N.; Savalgi, V. P. Effect of nitrogen fixing paenibacillus sp. Isolates on growth, yield and nutrient uptake on maize in calcareous soil. International Journal of Agricultural and Statistical Sciences. 14, pp. 427 - 431. DR RAM KISHAN, 01/01/2018. ISSN 0973-1903, ISSN 0976-3392
Márquez-Benavidez, L., Rizo-León, M. Á., Montaño-Arias, N. M., Ruiz-Nájera, R., & Sánchez-Yáñez, J. M. (2017). Respuesta de Phaseolus vulgaris a la inoculación de diferentes dosis de Trichoderma harzianum con el fertilizante nitrogenado reducido al 50%. Journal of the Selva Andina Research Society, 8(2), 135-144.
•    The discussion section also need some improvements with statements included previously on results section. 
It was changed in different lines. In results we moved lines to a discussion and now they are 473-503.At the same time, we added the lines 451-455 and 511-522.

Reviewer 2 Report

The manuscript entitled “Organic substrates and biofertilization in soilless crops and their effects on yield" is based on original research experiment and the presented results therein broaden the knowledge in the field of applied plant science and horticultural science. Authors aimed to evaluate the effects of both organic substrates in different mix (V+CF) and biofertilizers (PGPMs) on yield, drain age, and plant nutritional status (xylem sap) of two organic crops (Cucumis melo L. and Solanum lycopersicum L.). The scope of work includes the performance of experiment in controlled conditions, during which measurements of crop yield, nutrition and fertigation and dehydrogenase activity (DHA) were obtained.

There is no doubt that this work is in the scope of Horticulturae journal. The publication presents generally interesting and important studies. The paper is well organized, presented in a logical sequence, and has adequate bibliographic review (here I have one comment which I will list below). The work delivers interesting results and can be the important source of valuable information.

The introduction is properly composed. The materials and methods section contains the basic requested elements and provide information about the experimental preparations, analyses and growth conditions. The data analysis is properly provided. The results show valuable information. The obtained data are discussed sufficiently.

However, authors made some shortcomings that must be corrected before the publication of the work:

  • Introduction: While I have no objections to the aims of the work, in the same paragraph I am missing a scientific hypothesis.
  • Introduction: plant species should be characterized, also their economic importance.
  • MM section: where does the plant material come from?
  • MM section: Are the authors convinced that a ANOVA can be applied to this data? It is also not mentioned whether the normality of the distribution was tested.
  • MM section: subsection 2.5. does not need to be further divided into subsubsections.

Moreover, I suggest one publication, which can enrich the Introduction or Discussion section:

Vozhehova R., Kokovikhin S., Lykhovyd P. V., Balashova H., Lavrynenko Y., Biliaieva I., Markovska O. 2020. Statistical yielding models of some irrigated vegetable crops in dependence on water use and heat supply. Journal of Water and Land Development. No. 45 (IV–VI) p. 190–197. DOI: 10.24425/jwld.2020.133494.

I would like to underline that my remarks are auxiliary and not undertake the quality and importance of the paper.

Author Response

Thank you so much about the review´s commentaries. We tried to modification all of them. We hope to do it all and added higher quality to the article.

The correction of the work:
•    Introduction: While I have no objections to the aims of the work, in the same paragraph I am missing a scientific hypothesis.
The paper was editing in different lines, and at the same time we added some recommendations of reviewer 1. The changes:
o    The abstract lines are 23-25
o    Introduction lines are 40-45 and the moved information are 63-119.
o    Added a hypothesis in lines 120-125.
•    Introduction: plant species should be characterized, also their economic importance.
We thought which it wasn’t necessary due to fact that in this section but add another information about PGPM and substrates lines 63-119. Nonetheless, it was added in lines 219-228 with information about the culture in materials and methods. 
•    MM section: where does the plant material come from?
The plant material like C. melo is from HM Clause seeds and S. lycopersicum is from Rijk Zwaan seeds, this information is in the line 222-229. At the same time, Material and methods were improved with some recommendations of reviewer 1. In the case of experimental, was corrected a mistake in line 134 and added more information about that in line 134-137. About the control level the vermicompost, it was changed reference previous research for others in line (143). These references are:
Prasad, M.; Maher, M. J. The use of composted green waste (CGW) as a growing medium component. Acta Hortic. 2001, 549, 107–114.

Truong, H. D., & Wang, C. H. (2015). Studies on the effects of vermicompost on physicochemical properties and growth of two tomato varieties under greenhouse conditions. Communications in Soil Science and Plant Analysis, 46(12), 1494-1506. 
The drainage test information is added in lines 238-242.
•    MM section: Are the authors convinced that an ANOVA can be applied to this data? It is also not mentioned whether the normality of the distribution was tested.
Yes, we are convinced that the anova can be applied. The Multifactorial ANOVA procedure is designed to build a statistical model describing the impact of two or more categorical factors Xj (substrates, PGPMs) on a dependent variable Y (yield). Tests are performed to determine whether or not there are significant differences between the means at different levels of the factors and whether or not there are interactions between the factors. In addition, data can be displayed graphically in a number of ways, including a multiple scatter plot, a mean plot, and an interaction graph. It is also not mentioned whether the normality of the distribution was tested. Now it is mentioned in the line 278-281. 
•    MM section: subsection 2.5. does not need to be further divided into subsubsections.
The subsection 2.5. it was removed, and change for enumeration list. We added lines 247-248, 254-256 and 278-281.
•    The suggestion of Vozhehova et al., 2020.
Thank you so much for your suggestion, we used for the discussion in lines 451-455.
•     The discussion 
At the same time, we added some recommendations of reviewer 1. The discussion it was changed and added different lines. In results we moved lines to a discussion and now they are 473-503. Also, we added the lines 451-455 and 511-522.

Reviewer 3 Report

Comments are given in the file attatched. 

Reviewer 4 Report

Very nice work. I really enjoyed the approach you have taken.

Round 2

Reviewer 1 Report

The experimental work presented in the Manuscript horticulturae-1559742, entitled "Organic substrates and biofertilization in soilless crops and their effects on yield" is interesting research with promising results related with the utilization of organic materials for growing media and microorganisms as biofertilizers for soilless crops. The article reports the effects of both substrates and microorganism on several variables and indicators .  Several changes were included according to the comments of reviewers and the manuscript was improved.  Nevertheless, there are still some shortcomings that should be modified in order to enhance the final manuscript for the readers. Additionally, English need some editing again. 

Material and methods also section need some improvements 
-    In table 1 could you include physicochemical characterization of initial mixtures (treatments)
-    In terms of design, there two controls one for substrate and one for microorganisms. It is very confusing in the description of results. Please modify.
-    Please provide details about how substrates were produced or if were commercial indicate manufacturer antecedents.
-    Include details and references for vermicompost tea. How the soluble fraction was obtained and isolated for irrigation.
-    Related to biofertilizers, please provide antecedents how the microorganisms were isolated if correspond or if the inoculums were commercial products. Include details of manufacturer if correspond. 
-    Please provide antecedents of the inoculation process. Despite references were provide, the inoculation of AMF was performed by the addition of soil? Please include details.
-    Please include references for the quantification of microorganism’s procedure (lines 178-190)
-    In table 2, where the total microbial loads were described, please indicate which substrate or mixture (SUS) was considered for this analysis. 
-    In subsection 2.3, please indicate if the cultivation for both tomato and melon were carried out in containers as pots or similar, and if the total process was totally completed in soilless system.
-    In 231 to 248 indicate how leaching were collected. 
-    In parameters evaluated subsection, avoid, if possible, bullet points.

Results section
-    Despite there are several promising results, the description needs to be improved to facilitate the understanding of readers, it can be confusing. In this sense, I suggest modifying figures 1 and 2. Figure 1 would be illustrative by using histograms or bars and separated by factors showing what the letters represent. 
-    In case of figures 2 and 3, it would more informative if authors provide tables for crops similar to table 3 with the factors separated (PGPR instead DAT).
-    In lines 329 and 330 authors indicate that higher proportion of vermicompost and TRICH and PGPR showed the higher concentration of NH4 and K. However, in melon crop, control treatment without microorganisms represents the higher level of K and tomato shows no differences for NH4 and K between control and PGPR treatments.
-    In line 324 to 326 Authors argues that CF shows higher concentration of Na and Mg because its sorption capacity. However, the reference provided is related to activated CF that have another conditioned properties associated to the activation process. 
-    Lines 378 to 385 and lines 400 to 405 should be moved to discussion section. 
-    In description of PCA figures (lines 414 to 428), indicate which is for melon and tomato in the text. 

Conclusion section
-    Lines 548 to 553 should be avoid, considering the scope of the study. In this sense, no treatments with conventional fertilization were included under this research conditions. 

Author Response

Changes from February 8, 2022 
Comments and Changes from Authors
Dear Review:
We greatly appreciate your feedback. The article was modified some shortcomings to improve the manuscript.

Introduction 
Response: It was modified a sap test and reference line 82-84. 
Material and methods 
-    In table 1 could you include physicochemical characterization of initial mixtures (treatments)
Response: It was included and modified the table in line 148-149. 
-    In terms of design, there two controls one for substrate and one for microorganisms. It is very confusing in the description of results. Please modify.
Response: It was modified in line 138 for substrate and in line 142 for microorganisms.
-    Please provide details about how substrates were produced or if were commercial indicate manufacturer antecedents.
Response: This information it was included in line 146-147
-    Include details and references for vermicompost tea. How the soluble fraction was obtained and isolated for irrigation.
Response: This information it was included in line 243-250.
-    Related to biofertilizers, please provide antecedents how the microorganisms were isolated if correspond or if the inoculums were commercial products. Include details of manufacturer if correspond. 
Response: The antecedents how the microorganisms were isolate, and the details of manufacturer are included in the line 160-162, 167-169 and 173-174.

-    Please provide antecedents of the inoculation process. Despite references were provide, the inoculation of AMF was performed by the addition of soil? Please include details.
Response: The information is in line 176-182.
-    Please include references for the quantification of microorganism’s procedure (lines 178-190)
Response: Now is lines 187-196 and we included reference number 35.
López-González, J.A., López, M.J., Vargas-García, M.C., Suárez-Estrella, F., Jurado, M. y Moreno, J. Tracking organic matter and microbiota dynamics during the stages of lignocellulosic waste composting. Bioresource Technology 2013; 146, 574-584
-    In table 2, where the total microbial loads were described, please indicate which substrate or mixture (SUS) was considered for this analysis. 
Response: In line 214 is indicated the type of substrate (40V60CF). 
-    In subsection 2.3, please indicate if the cultivation for both tomato and melon were carried out in containers as pots or similar, and if the total process was totally completed in soilless system.
Response: The information about the containers  was included in line 227-228 and about total process was included in line 233.

-    In 231 to 248 indicate how leaching were collected. 
Response: This information is included in line 254-255.
-    In parameters evaluated subsection, avoid, if possible, bullet points.
Response: The bullet points were avoided in parameters evaluated subsection (lines 259-291).
Results section
-    Despite there are several promising results, the description needs to be improved to facilitate the understanding of readers, it can be confusing. In this sense, I suggest modifying figures 1 and 2. Figure 1 would be illustrative by using histograms or bars and separated by factors showing what the letters represent.
 Response: The figures 1 a and b modified and changed for bars graphics in line 307.
-    In case of figures 2 and 3, it would more informative if authors provide tables for crops similar to table 3 with the factors separated (PGPR instead DAT).
Response: The figures 2 and 3 were changed for tables in lines 325-332.
-    In lines 329 and 330 authors indicate that higher proportion of vermicompost and TRICH and PGPR showed the higher concentration of NH4 and K. However, in melon crop, control treatment without microorganisms represents the higher level of K and tomato shows no differences for NH4 and K between control and PGPR treatments.
Response: The information was added lines 335-343. 
-    In line 324 to 326 Authors argues that CF shows higher concentration of Na and Mg because its sorption capacity. However, the reference provided is related to activated CF that have another conditioned properties associated to the activation process. 
Response: These lines and reference were removed .

  • -    Lines 378 to 385 and lines 400 to 405 should be moved to discussion section.
     Response: These lines were moved to discussion section and now they are 540-550.
    -    In description of PCA figures (lines 414 to 428), indicate which is for melon and tomato in the text. 
    Response: The information was included line 411, 421  and 433. 
    Conclusion section
    -    Lines 548 to 553 should be avoid, considering the scope of the study. In this sense, no treatments with conventional fertilization were included under this research conditions. 
    Response: These lines were removed like reviewer suggested.
